# PAC-Bayes Tree: weighted subtrees with guarantees

**Tin Nguyen**[*]
MIT EECS
tdn@mit.edu

**Samory Kpotufe**
Princeton University ORFE
samory@princeton.edu

## Abstract

We present a weighted-majority classification approach over subtrees of a fixed tree, which provably achieves excess-risk of the *same* order as the best tree-pruning. Furthermore, the computational efficiency of pruning is maintained at both training and testing time despite having to aggregate over an exponential number of subtrees. We believe this is the first subtree aggregation approach with such guarantees.

The guarantees are obtained via a simple combination of insights from PAC-Bayes theory, which we believe should be of independent interest, as it generically implies consistency for weighted-voting classifiers w.r.t. Bayes – while, in contrast, usual PAC-bayes approaches only establish consistency of *Gibbs* classifiers.

## 1 Introduction

Classification trees endure as popular tools in data analysis, offering both efficient prediction and interpretability – yet they remain hard to analyze in general. So far there are two main approaches with generalization guarantees: in both approaches, a large tree (possibly overfitting the data) is first obtained; one approach is then to *prune* back this tree down to a subtree[2] that generalizes better; the alternative approach is to combine all possible subtrees of the tree by *weighted* majority vote. Interestingly, while both approaches are competitive with other practical heuristics, it remains unclear whether the alternative of *weighting subtrees* enjoys the same strong generalization guarantees as pruning; in particular, no weighting scheme to date has been shown to be *statistically consistent*, let alone attain the same tight generalization rates (in terms of excess risk) as pruning approaches.

In this work, we consider a new weighting scheme based on PAC-Bayesian insights [1], that (a) is consistent and attains the same generalization rates as the best pruning of a tree, (b) is efficiently computable at both training and testing time, and (c) competes against pruning approaches on real-world data. To the best of our knowledge, this is the first practical scheme with such guarantees.

The main technical hurdle has to do with a subtle tension between goals (a) and (b) above. Namely, let $T_0$ denote a large tree built on $n$ datapoints, usually a binary tree with $O(n)$ nodes; the family of subtrees $T$ of $T_0$ is typically of exponential size in $n$ [2], so a naive voting scheme that requires visiting all subtrees is impractical; on the other hand it is known that if the weights decompose favorably over the leaves of $T$ (e.g., multiplicative over leaves) then efficient classification is possible. Unfortunately, while various such multiplicative weights have been designed for voting with subtrees [3, 4, 5], they are not known to yield statistically consistent prediction. In fact, the best known result to date [5] presents a weighting scheme which can provably achieve an *excess* risk[3] (over the Bayes classifier) of the form $o_P(1) + C \cdot \min_T \mathcal{R}(h_T)$, where $\mathcal{R}(h_T)$ denotes the misclassification rate of a classifier $h_T$ based on subtree $T$. In other words, the excess risk might never go to 0 as sample size increases, which in contrast is a basic property of the pruning alternative. Furthermore, the approach

---

[*]The majority of the research was done when the author was an undergraduate student at Princeton University ORFE.

[2]Considering only subtrees that partition the data space.

[3]The excess risk of a classifier $h$ over the Bayes $h_B$ (which minimizes $\mathcal{R}(h)$ over any $h$) is $\mathcal{R}(h) - \mathcal{R}(h_B)$.

of [5], based on $l_1$-risk minimization, does not trivially extend to multiclass classification, which is most common in practice. Our approach is designed for multiclass by default.

**Statistical contribution.** PAC-Bayesian theory [1, 6, 7, 8] offers useful insights into designing weighting schemes with generalization guarantees (w.r.t. a *prior* distribution $P$ over classifiers). However, a direct application of existing results fails to yield a consistent weighted-majority scheme. This is because PAC-Bayes results are primarily concerned with so-called *Gibbs classifiers*, which in our context corresponds to predicting with a random classifier $h_T$ drawn according to a weight-distribution $Q$ over subtrees of $T_0$. Instead, we are interested in $Q$-weighted majority classifiers $h_Q$. Unfortunately the corresponding error $\mathcal{R}(h_Q)$ can be twice the risk $\mathcal{R}(Q) = \mathbb{E}_{h_T \sim Q} \mathcal{R}(h_T)$ of the corresponding Gibbs classifier: this then results (at best – see overview in Section 2.2) in an excess risk of the form $(\mathcal{R}(h_Q) - \mathcal{R}(h_B)) \leq (\mathcal{R}(Q) - \mathcal{R}(h_B)) + \mathcal{R}(Q) = o_P(1) + \mathcal{R}(h_B)$, which, similar to [5], does not go to $0$. So far, this problem is best addressed in PAC-Bayes results such as the MinCq bound in [6, 8] on $\mathcal{R}(h_Q)$, which is tighter in the presence of low correlation between base classifiers. In contrast, our PAC-Bayes result applies even without low correlation between base classifiers, and allows an excess risk $o_P(1) + (C/n) \cdot \min_T \log(1/P(T)) \to 0$ (Proposition 2). This first result is in fact of general interest since it extends beyond subtrees to any family of classifiers, and is obtained by carefully combining existing arguments from PAC-Bayes analysis.

However, our basic PAC-Bayes result alone does not ensure convergence at the same rate as that of the best pruning approaches. This requires designing a prior $P$ that scales properly with the size of subtrees $T$ of $T_0$. For instance, suppose $P$ were uniform over all subtrees of $T_0$, then $\log(1/(P(T)) = \Omega(n)$, yielding a vacuous excess risk. We show through information-theoretic arguments that an appropriate prior $P$ can be designed to yield rates of convergence of the same order as that of the best pruning of $T_0$. In particular, our resulting weighting scheme maintains ideal properties of pruning approaches such as *adaptivity* to the intrinsic dimension of data (see e.g. [9]).

**Algorithmic contribution.** We show that we can design a prior $P$ which, while meeting the above statistical constraints, yields *posterior* weights that decompose favorably over the leaves of a subtree $T$. As a result of this decomposition, the weights of all subtrees can be recovered by simply maintaining corresponding weights at the nodes of the original tree $T_0$ for efficient classification in time $O(\log n)$ (this is illustrated in Figure 1). We then propose an efficient approach to obtain weights at the nodes of $T_0$, consisting of concurrent top-down and bottom-up dynamic programs that run in $O(n)$ time. These match the algorithmic complexity of the most efficient pruning approaches, and thus offer a practical alternative.

Our theoretical results are then verified in experiments over many real-world datasets. In particular we show that our weighted-voting scheme achieves similar or better error than pruning on practical problems, as suggested by our theoretical results.

**Paper Organization.** We start in Section 2 with theoretical setup and an overview of PAC-Bayes analysis. This is followed in Section 3 with an overview of our statistical results, and in Section 4 with algorithmic results. Our experimental analysis is then presented in Section 5.

## 2 Preliminaries

### 2.1 Classification setup

We consider a multiclass setup where the input $X \subset \mathcal{X}$, for a bounded subset $\mathcal{X}$ of $\mathbb{R}^D$, possibly of lower intrinsic dimension. For simplicity of presentation we assume $\mathcal{X} \subset [0, 1]^D$ (as in normalized data). The output $Y \subset [L]$, where we use the notation $[L] = \{1, 2, \ldots, L\}$ for $L \in \mathbb{N}$.

We are to learn a *classifier* $h : \mathcal{X} \mapsto [L]$, given an i.i.d. training sample $\{X_i, Y_i\}_{i=1}^{2n}$ of size $2n$, from an unknown distribution over $X, Y$. Throughout, we let $\mathcal{S} \doteq \{X_i, Y_i\}_{i=1}^{n}$ and $\mathcal{S}_0 \doteq \{X_i, Y_i\}_{i=n+1}^{2n}$, which will serve later to simplify dependencies in our analysis.

Our performance measure is as follows.

**Definition 1.** *The* **risk** *of a classifier $h$ is given as $\mathcal{R}(h) = \mathbb{E}[h(X) \neq Y]$. This is minimized by the* Bayes *classifier $h_B(x) \doteq \mathrm{argmax}_{l \in [L]} \mathbb{P}(Y = l | X = x)$. Therefore, for any classifier $\hat{h}$ learned over a sample $\{X_i, Y_i\}_i$, we are interested in the* **excess-risk** *$\mathcal{E}(\hat{h}) \doteq \mathcal{R}(\hat{h}) - \mathcal{R}(h_B)$.*

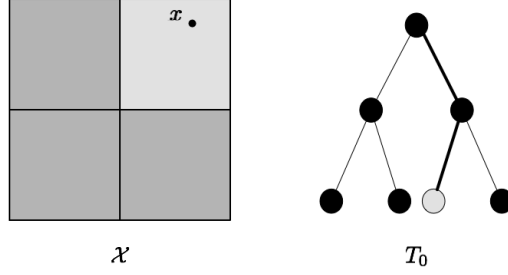

$$\mathcal{X} \qquad\qquad T_0$$

Figure 1: A partition tree $T_0$ over input space $\mathcal{X}$, and a query $x \in \mathcal{X}$ to classify. The leaves of $T_0$ are the 4 cells shown left, and the root is $\mathcal{X}$. A query $x$ follows a single path (shown in bold) from the root down to a leaf. A key insight towards efficient weighted-voting is that this path visits all leaves (containing $x$) of any subtree of $T_0$. Therefore, weighted voting might be implemented by keeping a weight $w(A)$ at any node $A$ along the path, where $w(A)$ aggregates the weights $Q(T)$ of every subtree $T$ that has $A$ as a leaf. This is feasible if we can restrict $Q(T)$ to be multiplicative over the leaves of $T$, without trading off accuracy.

Here we are interested in aggregations of *classification trees*, defined as follows.

**Definition 2.** *A hierarchical partition or (space) partition-tree $T$ of $\mathcal{X}$ is a collection of nested partitions of $\mathcal{X}$; this is viewed as a tree where each node is a subset $A$ of $\mathcal{X}$, each child $A'$ of a node $A$ is a subset of $A$, and whose collection of leaves, denoted $\pi(T)$, is a partition of $\mathcal{X}$. A* **classification tree** *$h_T$ on $\mathcal{X}$ is a labeled partition-tree $T$ of $\mathcal{X}$: each leaf $A \in \pi(T)$ is assigned a label $l = l(A) \in [L]$; the classification rule is simply $h_T(x) = l(A)$ for any $x \in A$.*

*Given an initial tree $T_0$, we will consider only* **subtrees** *$T$ of $T_0$ that form a hierarchical partition of $\mathcal{X}$, and we henceforth use the term* subtrees *(of $T_0$) without additional qualification.*

Finally, aggregation (of subtrees of $T_0$) consists of *majority-voting* as defined below.

**Definition 3.** *Let $\mathcal{H}$ denote a discrete family of classifiers $h : \mathcal{X} \mapsto [L]$, and let $Q$ denote a distribution over $\mathcal{H}$. The $Q$-**majority** classifier $h_Q \doteq h_Q(\mathcal{H})$ is one satisfying for any $x \in \mathcal{X}$*

$$h_Q(x) = \operatorname*{argmax}_{l \in [L]} \sum_{h \in \mathcal{H}, h(x)=l} Q(h).$$

Our oracle rates of Theorem 1 requires no additional assumptions; however, the resulting corollary is stated under standard distributional conditions that characterize convergence rates for tree-prunings.

## 2.2 PAC-Bayes Overview

PAC-Bayes analysis develops tools to bound the error of a Gibbs classifier, i.e. one that randomly samples a classifier $h \sim Q$ over a family of classifiers $\mathcal{H}$. In this work we are interested in families $\{h_T\}$ defined over subtrees of an initial tree $T_0$. Here we present some basic PAC-Bayes result which we extend for our analysis. While these results are generally presented for classification risk $\mathcal{R}$ (defined above), we keep our presentation generic, as we show later that a different choice of risk leads to stronger results for $\mathcal{R}$ than what is possible through direct application of existing results.

**Generic Setup.** Consider a random vector $Z$, and an i.i.d sample $Z_{[n]} = \{Z_i\}_{i=1}^n$. Let $\mathcal{Z}$ be the support of $Z$, and $\mathcal{L} = \{\ell_h : h \in \mathcal{H}\}$ be a *loss class* indexed by $h \in \mathcal{H}$ – *discrete*, and where $\ell_h : \mathcal{Z} \to [0, 1]$. For $h \in \mathcal{H}$, the loss $\ell_h$ induces the following risk and empirical counterparts:

$$\mathcal{R}_{\mathcal{L}}(h) \doteq \mathbb{E}_Z \ell_h(Z), \quad \widehat{\mathcal{R}}_{\mathcal{L}}(h, Z_{[n]}) \doteq \frac{1}{n} \sum_{i=1}^n \ell_h(Z_i).$$

In particular, for the above classification risk $\mathcal{R}$, and $Z \triangleq (X, Y)$, we have $\ell_h(Z) = \mathbb{1}\{h(X) \neq Y\}$.

Given a distribution $Q$ over $\mathcal{H}$, the risk (and empirical counterpart) of the Gibbs classifier is then

$$\mathcal{R}_{\mathcal{L}}(Q) \doteq \mathbb{E}_{h \sim Q} \mathcal{R}_{\mathcal{L}}(h), \quad \widehat{\mathcal{R}}_{\mathcal{L}}(Q, Z_{[n]}) \doteq \mathbb{E}_{h \sim Q} \widehat{\mathcal{R}}_{\mathcal{L}}(h, Z_{[n]}).$$

PAC-Bayesian results bound $\mathcal{R}_{\mathcal{L}}(Q)$ in terms of $\widehat{\mathcal{R}}_{\mathcal{L}}(Q, Z_{[n]})$, *uniformly* over any distribution $Q$, provided a fixed *prior* distribution $P$ over $\mathcal{H}$. We will build on the following form of [10] which yields an upper-bound that is convex in $Q$ (and therefore can be optimized for a good *posterior* $Q^*$).

**Proposition 1** (PAC-Bayes on $\mathcal{R}_{\mathcal{L}}$ [10]). *Fix a prior $P$ supported on $\mathcal{H}$, and let $n \geq 8$ and $\delta \in (0, 1)$. With probability at least $1 - \delta$ over $Z_{[n]}$, simultaneously for all $\lambda \in (0, 2)$ and all posteriors $Q$ over $\mathcal{H}$:*

$$\mathcal{R}_{\mathcal{L}}(Q) \leq \frac{\widehat{\mathcal{R}}_{\mathcal{L}}(Q, Z_{[n]})}{1 - \lambda/2} + \frac{\mathcal{D}_{kl}(Q\|P) + \log(2\sqrt{n}/\delta)}{\lambda(1 - \lambda/2)n},$$

*where $\mathcal{D}_{kl}(Q\|P) \doteq \mathbb{E}_Q \log \frac{Q(h)}{P(h)}$ is the Kullback-Leibler divergence between $Q$ and $P$.*

**Choice of posterior $Q^*$.** Let $Q^*$ minimize the above upper-bound, and let $h^*$ minimize $\mathcal{R}_{\mathcal{L}}$ over $\mathcal{H}$. Then, by letting $Q_{h^*}$ put all mass on $h^*$, we automatically get that, with probability at least $1 - 2\delta$:

$$\mathcal{R}_{\mathcal{L}}(Q^*) \leq \mathcal{R}_{\mathcal{L}}(Q_{h^*}) \leq C \cdot \left( \widehat{\mathcal{R}}_{\mathcal{L}}(h^*, Z_{[n]}) + \frac{\log(1/P(h^*)) + \log(n/\delta)}{n} \right)$$

$$\leq C \cdot \left( \mathcal{R}_{\mathcal{L}}(h^*) + \frac{\log(1/P(h^*)) + \log(n/\delta)}{n} + \sqrt{\frac{\log(1/\delta)}{n}} \right), \tag{1}$$

where the last inequality results from bounding $|\mathcal{R}_{\mathcal{L}}(h^*) - \widehat{\mathcal{R}}_{\mathcal{L}}(h^*, Z_{[n]})|$ using Chernoff.

Unfortunately, such direct application is not enough for our purpose when $\mathcal{R}_{\mathcal{L}} = \mathcal{R}$. We want to bound the excess risk $\mathcal{E}(h_Q)$ for a $Q$-majority classifier $h_Q$ over $h's \in \mathcal{H}$. It is known that $\mathcal{R}(h_Q) \leq 2\mathcal{R}(Q)$ which yields a bound of the form (1) on $\mathcal{R}(h_{Q^*})$; however this implies at best that $\mathcal{R}(h_{Q^*}) \to 2\mathcal{R}(h_B)$ even if $\mathcal{E}(h^*) \to 0$ (which is generally the case for optimal tree-pruning $h_T^*$ [9]). This is a general problem in converting from Gibbs error to that of majority-voting, and is studied for instance in [6, 8] where it is shown that $\mathcal{R}(h_Q)$ can actually be smaller in some situations.

**Improved choice of $Q^*$.** Here, we want to design $Q^*$ such that $\mathcal{R}(h_{Q^*}) \to \mathcal{R}(h_B)$ (i.e. $\mathcal{E}(h_{Q^*}) \to 0$) at the same rate as $\mathcal{E}(h_T^*) \to 0$ always. Our solution relies on a proper choice of loss $\ell_h$ that relates most directly to excess risk $\mathcal{E}$ that the 0-1 loss $\mathbb{1}\{h(x) \neq y\}$. A first candidate is to define $\ell_h(x, y)$ as $e_h(x, y) \doteq \mathbb{1}\{h(x) \neq y\} - \mathbb{1}\{h_B(x) \neq y\}$ since $\mathcal{E}(h) = \mathbb{E} e_h(X, Y)$; however $e_h(x, y) \notin [0, 1]$ and can take negative values. This is resolved by considering an intermediate loss $e_h(x) = \mathbb{E}_{Y|x} e_h(x, Y) \in [0, 1]$ to be related back to $e_h(x, y)$ by integration in a suitable order.

## 3 Statistical results

### 3.1 Basic PAC-Bayes result

We start with the following intermediate loss family over classifiers $h$, w.r.t. the Bayes classifier $h_B$.

**Definition 4.** *Let $e_h(x, y) \doteq \mathbb{1}\{h(x) \neq y\} - \mathbb{1}\{h_B(x) \neq y\}$, and $e_h(x) = \mathbb{E}_{Y|x} e_h(x, Y)$, and*

$$\widetilde{\mathcal{E}}(h, \mathcal{S}) \doteq \frac{1}{n} \sum_{i=1}^{n} e_h(X_i), \text{ and } \widehat{\mathcal{E}}(h, \mathcal{S}) \doteq \frac{1}{n} \sum_{i=1}^{n} e_h(X_i, Y_i).$$

Our first contribution is a basic PAC-Bayes result which the rest of our analysis builds on.

**Proposition 2** (PAC-Bayes on excess risk). *Let $\mathcal{H}$ denote a discrete family of classifiers, and fix a* prior *distribution $P$ with support $\mathcal{H}$. Let $n \geq 8$ and $\delta \in (0, 1)$. Suppose, there exists bounded functions $\widehat{\Delta}_n(h, \mathcal{S}), \Delta_n(h), h \in \mathcal{H}$ (depending on $\delta$) such that*

$$\mathbb{P}\left(\forall h \in \mathcal{H}, \widetilde{\mathcal{E}}(h, \mathcal{S}) \leq \widehat{\mathcal{E}}(h, \mathcal{S}) + \widehat{\Delta}_n(h, \mathcal{S})\right) \geq 1 - \delta, \quad \inf_{h \in \mathcal{H}} \mathbb{P}\left(\widehat{\Delta}_n(h, \mathcal{S}) \leq \Delta_n(h)\right) \geq 1 - \delta.$$

*For any $\lambda \in (0, 2)$, consider the following posterior over $\mathcal{H}$:*

$$Q_\lambda^*(h) = \frac{1}{c} e^{-n\lambda(\widehat{\mathcal{R}}(h, \mathcal{S}) + \widehat{\Delta}_n(h, \mathcal{S}))} P(h), \quad \text{for } c = \mathbb{E}_{h \sim P} e^{-n\lambda(\widehat{\mathcal{R}}(h, \mathcal{S}) + \widehat{\Delta}_n(h, \mathcal{S}))}. \tag{2}$$

*Then, with probability at least $1 - 4\delta$ over $\mathcal{S}$, simultaneously for all $\lambda \in (0, 2)$:*

$$\mathcal{E}(h_{Q_\lambda^*}) \leq \frac{L}{1 - \lambda/2} \inf_{h \in \mathcal{H}} \left( \mathcal{E}(h) + \Delta_n(h) + \frac{\log(1/P(h))}{\lambda n} + \frac{\log \frac{2\sqrt{n}}{\delta} + \lambda\sqrt{2n \log \frac{1}{\delta}}}{\lambda n} \right).$$

Proposition 2 builds on Proposition 1 by first taking $\mathcal{R}_{\mathcal{L}}(h)$ to be $\mathcal{E}(h)$, $\widehat{\mathcal{R}}_{\mathcal{L}}(h)$ to be $\widetilde{\mathcal{E}}(h)$, and $Z$ to be $X$. The bound in Proposition 2 is then obtained by optimizing over $Q$ for fixed $\lambda$. Since this bound is on excess error (rather than error), optimizing over $\lambda$ can only improve constants, while the choice of prior $P$ is crucial in obtaining optimal rates as $|\mathcal{H}| \to \infty$. Such choice is treated next.

## 3.2 Oracle risk for trees (as $\mathcal{H} \doteq \mathcal{H}(T_0)$ grows in size with $T_0$)

We start with the following definitions on classifiers of interest and related quantities.

**Definition 5.** *Let $T_0$ be a binary partition-tree of $\mathcal{X}$ obtained from data $\mathcal{S}_0$, of depth $D_0$. Consider a family of classification trees $\mathcal{H}(T_0) \doteq \{h_T\}$ indexed by subtrees $T$ of $T_0$, and where $h_T$ defines a fixed labeling $l(A)$ of nodes $A \in \pi(T)$, e.g., $l(A) \doteq$ majority label in $Y$ if $A \cap \mathcal{S}_0 \neq \emptyset$.*

*Furthermore, for any node $A$ of $T_0$, let $\hat{p}(A, \mathcal{S})$ denote the empirical mass of $A$ under $\mathcal{S}$ and $p(A)$ be the population mass. Then for any subtree $T$ of $T_0$, let $|T|$ be the number of nodes in $T$ and define*

$$\widehat{\Delta}_n(h_T, \mathcal{S}) \doteq \sum_{A \in \pi(T)} \sqrt{\hat{p}(A, \mathcal{S}) \frac{2 \log(|T_0| / \delta)}{n}}, \, and \tag{3}$$

$$\Delta_n(h_T) \doteq \sum_{A \in \pi(T)} \sqrt{8 \max \left( p(A), \frac{(2 + \log D) \cdot D_0 + \log(1/\delta)}{n} \right) \frac{\log(|T_0| / \delta)}{n}}. \tag{4}$$

**Remark 1.** *In practice, we might start with a space partitioning tree $T_0'$ (e.g., a dyadic tree, or KD-tree) which partitions $[0,1]^D$, rather than the support $\mathcal{X}$. We then view $T_0$ as the intersection of $T_0'$ with $\mathcal{X}$.*

Our main theorem below follows from Proposition 2 on excess risk, by showing (a) that the above definition of $\widehat{\Delta}_n(h_T, \mathcal{S})$ and $\Delta_n(h_T)$ satisfies the conditions of Proposition 2, and (b) that there exists a proper prior $P$ such that $\log(1/P(T)) \sim |\pi(T)|$, i.e., depends just on the subtree complexity rather than on that of $T_0$. The main technicality in showing (b) stems from the fact that $P$ needs to be a proper distribution (i.e. $\sum_T P(T) = 1$) without requiring too large a normalization constant (remember that the number of subtrees can be exponential in the size of $T_0$). This is established through arguments from coding theory, and in particular Kraft-McMillan inequality.

**Theorem 1** (Oracle risk for trees). *Let the prior satisfy $P(h_T) \doteq (1/C_P) e^{-3D_0 \cdot |\pi(T)|}$ for a normalizing constant $C_P$, and consider the corresponding posterior $Q_\lambda^*$ as defined in Equation 2, such that, with probability at least $1 - 4\delta$ over $\mathcal{S}$, for all $\lambda \in (0, 2)$, the excess risk $\mathcal{E}(h_{Q_\lambda^*})$ of the majority-classifier is at most*

$$\left(\frac{L}{1 - \lambda/2}\right) \cdot \min_{h_T \in \mathcal{H}(T_0)} \left( \mathcal{E}(h_T) + \Delta_n(h_T) + \frac{3D_0 \cdot |\pi(T)|}{\lambda n} + \frac{\log \frac{2\sqrt{n}}{\delta} + \lambda \sqrt{2n \log \frac{1}{\delta}}}{\lambda n} \right).$$

From Theorem 1 we can deduce that the majority classifier $h_{Q_\lambda^*}$ is consistent whenever the approach of pruning to the best subtree is consistent (typically, $\min_{h_T} \mathcal{E}(h_T) + (D_0 |\pi(T)|)/n = o_P(1)$). Furthermore, we can infer that $\mathcal{E}(h_{Q_\lambda^*})$ converges at the same rate as pruning approaches: the terms $\Delta_n(h_T)$ and $D_0 \cdot |\pi(T)|/n$ can be shown to be typically, of lower or similar order as $\mathcal{E}(h_T)$ for the best subtree classifier $h_T$. These remarks are formalized next and result in Corollary 1 below.

## 3.3 Rate of convergence

Much of known rates for tree-pruning are established for dyadic trees (see e.g. [9, 11]), due to their simplicity, under nonparametric assumptions on $\mathbb{E}[Y|X]$. Thus, we adopt such standard assumptions here to illustrate the rates achievable by $h_{Q_\lambda^*}$, following the more general statement of Theorem 1.

The first standard assumption below restricts how fast class probabilities change over space.

**Assumption 1.** *Consider the so-called **regression function** $\eta(x) \in \mathbb{R}^L$ with coordinate $\eta_l(x) \doteq \mathbb{E}_{Y|x} \mathbb{1}\{Y = l\}, l \in [L]$. We assume $\eta$ is $\alpha$-Hölder for $\alpha \in (0, 1]$, i.e.,*

$$\exists \lambda \text{ such that } \forall x, x' \in \mathcal{X}, \quad \|\eta(x) - \eta(x')\| \leq \lambda \|x - x'\|^\alpha.$$

Next, we illustrate some of the key conditions verified by dyadic trees which standard results build on. In particular, we want the *diameters* of nodes of $T_0$ to decrease relatively fast from the root down.

**Assumption 2** (Conditions on $T_0$). *The tree $T_0$ is obtained as the intersection of $\mathcal{X}$ with dyadic partition of $[0,1]^D$ (e.g. by cycling though coordinates) of depth $D_0 = O(D \log n)$ and partition size $|T_0| = O(n)$. In particular, we emphasize that the following conditions on subtrees then hold.*

*For any subtree $T$ of $T_0$, let $r(T)$ denote the maximum diameter of leaves of $T$ (viewed as subsets of $\mathcal{X}$). There exist $C_1, C_2, d > 0$ such that:*
*For all $(C_1/n) < r \leq 1$, there exists a subtree $T$ of $T_0$ such that $r(T) \leq r$ and $|\pi(T)| \leq C_2 r^{-d}$.*

The above conditions on subtrees are known to *approximately* hold for other procedures such as KD-trees, and PCA-trees; in this sense, analyses of dyadic trees do yield some insights into the performance other approaches. The quantity $d$ captures the *intrinsic dimension* (e.g., *doubling* or *box* dimension) of the data space $\mathcal{X}$ or is often of the same order [12, 13, 14].

Under the above two assumptions, it can be shown through standard arguments that the excess error of the best pruning, namely $\min_{h_T \in \mathcal{H}(T_0)} \mathcal{E}(h_T)$ is of order $n^{-\alpha/(2\alpha+d)}$, which is tight (see e.g. minimax lower-bounds of [15]). The following corollary to Theorem 1 states that such a rate, up to a logarithmic factor of $n$, is also attained by majority classification under $Q_\lambda^*$.

**Corollary 1** (Adaptive rate of convergence). *Assume that for any cell $A$ of $T_0$, the labeling $l(A)$ corresponds to the majority label in $A$ (under $\mathcal{S}_0$) if $A \cap \mathcal{S}_0 \neq \emptyset$, or $l(A) = 1$ otherwise. Then, under Assumptions 1 and 2, and the conditions of Theorem 1, there exists a constant $C$ such that:*

$$\mathbb{E}_{\mathcal{S}_0, \mathcal{S}} \mathcal{E}(h_{Q_\lambda^*}) \leq C \left( \frac{\log n}{n} \right)^{\alpha/(2\alpha+d)}.$$

## 4  Algorithmic Results

Here we show that $h_Q$ can be *efficiently* implemented by storing appropriate weights at nodes of $T_0$. Let $w_Q(A) \doteq \sum_{h_T : A \in \pi(T)} Q(h_T)$ aggregate weights over all subtrees $T$ of $T_0$ having $A$ as a leaf. Then $h_Q(x) = \operatorname{argmax}_{l \in [L]} \sum_{A \in \text{path}(x), l(A)=l} w_Q(A)$, where $\text{path}(x)$ denotes all nodes of $T_0$ containing $x$. Thus, $h_Q(x)$ is computable from weights proportional to $w_Q(A)$ at every node.

We show in what follows that we can efficiently obtain $w(A) = C \cdot w_{Q_\lambda^*}(A)$ by dynamic-programming by ensuring that $Q_\lambda^*(h_T)$ is multiplicative over $\pi(T)$. This is the case, given our choice of prior from Theorem 1: we have $Q_\lambda^*(h_T) = (1/C_{Q_\lambda^*}) \cdot \exp(\sum_{A \in \pi(T)} \phi(A))$ where

$$\phi(A) \doteq -\lambda \sum_{i : X_i \in A \cap \mathcal{S}} \mathbb{1}\{Y_i \neq l(A)\} - n\lambda \sqrt{\hat{p}(A, \mathcal{S}) \frac{2 \log(|T_0|/\delta)}{n}} - 3D_0.$$

We can then compute $w(A) \doteq C_{Q_\lambda^*} \cdot w_{Q_\lambda^*}(A)$ via dynamic-programming. The intuition is similar to that in [5], however, the particular form of our weights require a two-pass dynamic program (bottom-up and top-down) rather than the single pass in [5]. Namely, $w(A)$ divides into subweights that any node $A'$ might contribute up or down the tree. Let

$$\alpha(A) \doteq \sum_{h_T : A \in \pi(T)} \exp \left( \sum_{A' \neq A, A' \in \pi(T)} \phi(A') \right), \tag{5}$$

so that $w(A) = e^{\phi(A)} \cdot \alpha(A)$. As we will show (proof of Theorem 2), $\alpha(A)$ decomposes into contributions from the parent $A_p$ and sibling $A_s$ of $A$, i.e., $\alpha(A) = \alpha(A_p)\beta(A_s)$ where $\beta(A_s)$ is given as (writing $T_0^A$ for the subtree of $T_0$ rooted at $A$, and $T \preceq T'$ when $T$ is a subtree of $T'$):

$$\beta(A_s) = \sum_{T \preceq T_0^{A_s}} \exp \left( \sum_{A' \in \pi(T)} \phi(A') \right). \tag{6}$$

The contributions $\beta(A)$ are first computed using the bottom-up Algorithm 1, and the contributions $\alpha(A)$ and final weights $w(A)$ are then computed using the top-down Algorithm 2. For ease of presentation, these routines run on a full-binary tree version $\bar{T}_0$ of $T_0$, obtained by adding a dummy child to each node $A$ that has a single child in $T_0$. Each dummy node $A'$ has $\phi(A') = 0$.

**Algorithm 1** Bottom-up pass

---

**for** $A \in \pi(\bar{T}_0)$ **do**
    $\beta(A) \leftarrow e^{\phi(A)}$
**end for**
**for** $i \leftarrow D_0$ **to** $0$ **do**
    $\mathcal{A}_i \leftarrow$ set of nodes of $\bar{T}_0$ at depth $i$
    **for** $A \in \mathcal{A}_i \setminus \pi(\bar{T}_0)$ **do**
        $\mathcal{N} \leftarrow$ the children nodes of $A$
        $\beta(A) \leftarrow e^{\phi(A)} + \prod_{A' \in \mathcal{N}} \beta(A')$
    **end for**
**end for**

---

**Algorithm 2** Top-down pass

---

$\alpha(\text{root}) \leftarrow 1$
**for** $i \leftarrow 1$ **to** $D_0$ **do**
    $\mathcal{A}_i \leftarrow$ set of nodes of $\bar{T}_0$ at depth $i$
    **for** $A \in \mathcal{A}_i$ **do**
        $A_p, A_s \leftarrow$ parent of node $A$, sibling of node $A$
        $\alpha(A) \leftarrow \alpha(A_p)\beta(A_s)$
        $w(A) \leftarrow e^{\phi(A)}\alpha(A)$
    **end for**
**end for**

---

**Theorem 2** (Computing $w(A)$). *Running Algorithm 1, then 2, we obtain $w(A) \doteq C_{Q_\lambda^*} \cdot w_{Q_\lambda^*}(A)$, where $Q_\lambda^*$ is as defined in Theorem 1. Furthermore, the combined runtime of Algorithms 1, then 2 is $2|\bar{T}_0| \leq 4|T_0|$, where $|T|$ is the number of nodes in $T$.*

## 5 Experiments

Table 1: UCI datasets

| Name (abbreviation) | Features count | Labels count | Train size |
|---|---|---|---|
| Spambase (spam) | 57 | 2 | 2601 |
| EEG Eye State (eeg) | 14 | 2 | 12980 |
| Epileptic Seizure Recognition (epileptic) | 178 | 2 | 9500 |
| Crowdsourced Mapping (crowd) | 28 | 6 | 8546 |
| Wine Quality (wine) | 12 | 11 | 4497 |
| Optical Recognition of Handwritten Digits (digit) | 64 | 10 | 3620 |
| Letter Recognition (letter) | 16 | 26 | 18000 |

Here we present experiments on real-world datasets, for two common partition-tree approaches, dyadic trees and KD-trees. The various datasets are described in Table 1.

The **main baseline** we compare against, is a popular efficient pruning heuristic where a subtree of $T_0$ is selected to minimize the penalized error $\mathcal{C}_1(h_T) = \widehat{\mathcal{R}}(h_T, \mathcal{S}) + \lambda \frac{|\pi(T, \mathcal{S})|}{n}$.

We also compare against other tree-based approaches that are theoretically driven and efficient. First is a pruning approach proposed in [16], which picks a subtree minimizing the penalized error $\mathcal{C}_2(h_T) = \widehat{\mathcal{R}}(h_T, \mathcal{S}) + \lambda \sum_A \sqrt{\max\left(\hat{p}(A, \mathcal{S}), \frac{\|A\|}{n}\right) \cdot \frac{\|A\|}{n}}$, where $\|A\|$ denotes the depth of node $A$ in $T_0$. We note that, here we choose a form of $\mathcal{C}_2$ that avoids theoretical constants that were of a technical nature, but instead let $\lambda$ account for such. We report this approach as **SN-pruning**. Second is the majority classifier of [5], which however is geared towards binary classification as it requires *regression*-type estimates in $[0, 1]$ at each node. This is denoted **HS-vote**.

All the above approaches have efficient dynamic programs that run in time $O(|T_0|)$, and all predict in time $O(\text{height}(T_0))$. The same holds for our PAC-Bayes approach as discussed above in Section 4.

**Practical implementation of PAC-Bayes tree.** Our implementation rests on the theoretical insights of Theorem 1, however we avoid some of the technical details that were needed for rigor,

such as sample splitting and overly conservative constants in concentration results. Instead we advise cross-validating for such constants in the prior and posterior definitions. Namely, we first set $P(h_T) \propto \exp(-|\pi(T,\mathcal{S})|)$, where $\pi(T,\mathcal{S})$ denotes the leaves of $T$ containing data. We set $\Delta_n(h_T,\mathcal{S}) = \sum_{A \in \pi(T,\mathcal{S})} \sqrt{\frac{\hat{p}(A,\mathcal{S})}{n}}$. The posterior is then set as $Q^*(h_T) \propto \exp(-n(\lambda_1 \widehat{\mathcal{R}}(h_T,\mathcal{S}) + \lambda_2 \Delta_n(h_T,\mathcal{S})))P(h_T)$, where $\lambda_1, \lambda_2$ account for concentration terms to be tuned to the data.

Finally, we use the entire data to construct $T_0$ and compute weights, i.e., $\mathcal{S}_0 = \mathcal{S}$, as inter-dependencies are in fact less of an issue in practice. We note, that the above alternative theoretical approaches, SN-pruning and HS-vote, are also assumed (in theory) to work on a sample independent choice of $T_0$ (or equivalently built and labeled on a separate sample $S_0$), but are implemented here on the entire data to similarly take advantage of larger data sizes. The baseline pruning heuristic is by default always implemented on the full data.

**Experimental setup and results.** The data is preprocessed as follows: for dyadic trees, data is scaled to be in $[0,1]^D$, while for KD-trees data is normalized accross each coordinate by standard deviation.

Testing data is fixed to be of size 2000, while each experiment is ran 5 times (with random choice of training data of size reported in Table 1) and average performance is reported. In each experiment, all parameters are chosen by 2-fold cross-validation for each of the procedures. The log-grid is 10 values, equally spaced in logarithm, from $2^{-8}$ to $2^6$ while the linear-grid is 10 linearly-spaced values between half the best value of the log-search and twice the best value of the log-search.

Table 2 reports classification performance of the various theoretical methods relative to the baseline pruning heuristic. We see that proposed PAC-Bayes tree achieves competitive performance against all other alternatives. All the approaches have similar performance accross datasets, with some working slightly better on particular datasets. Figure 2 further illustrates typical performance on multiclass problems as training size varies.

Table 2: Ratio of classification error over that of the default pruning baseline: bold indicates best results across methods, while blue indicates improvement over baseline; N/A means the algorithm was not run on the task.

| Dataset | $T_0 \equiv$ dyadic tree | | | $T_0 \equiv$ KD tree | | |
|---|---|---|---|---|---|---|
| | SN-pruning | PAC-Bayes tree | HS-vote | SN-pruning | PAC-Bayes tree | HS-vote |
| spam | 1.118 | **0.975** | 1.224 | 1.048 | 1.020 | 1.075 |
| eeg | **0.979** | 0.993 | 1.029 | 1.000 | 0.990 | 1.000 |
| epileptic | 0.993 | 0.992 | 0.951 | 0.977 | 0.987 | **0.907** |
| crowd | **0.991** | 1.020 | N/A | 1.001 | 1.017 | N/A |
| wine | 1.035 | **0.991** | N/A | 1.010 | 0.997 | N/A |
| digit | 1.000 | **0.936** | N/A | 0.994 | 0.997 | N/A |
| letter | 1.005 | **0.993** | N/A | 1.000 | 1.001 | N/A |

Figure 2: Classification error versus training size

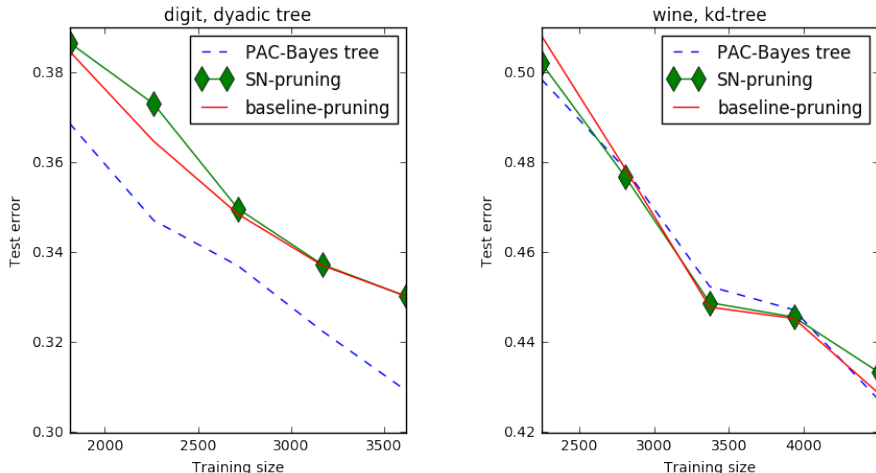

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
