[Supplementary Material]

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

# 6 Appendix

## 6.1 Proposition 2

The proof Proposition 2 requires the following lemma, which states that the excess risk of the voting classifier is always at most $L$ times the excess risk of the Gibbs classifier defined by the same distribution. It has the same spirit as the well-known result for the binary case where where the classification risk of the voting classifier is at most 2 times that of the stochastic classifier.

**Lemma 1.** *For any Q, it holds that:*

$$\mathcal{E}(h_Q) \leq L \cdot \mathcal{E}(Q).$$

*Proof.* For any classifier $h$, we have the following decomposition:

$$\mathcal{E}(h) = \mathbb{E}_X [\mathbb{P}(h_B(x) = Y | X = x) - \mathbb{P}(h(x) = Y | X = x)]$$

Interchanging order of integration, we write the excess risk of the Gibbs classifier as:

$$\mathcal{E}(Q) = \mathbb{E}_{h \sim Q} \mathcal{E}(h) = \mathbb{E}_X \left[ \mathbb{E}_{h \sim Q} \left( \mathbb{P}(Y = h_B(x) | X = x) - \mathbb{P}(Y = h(x) | X = x) \right) \right]$$

Similarly, the excess risk of the majority classifier is:

$$\mathcal{E}(h_Q) = \mathbb{E}_X [\mathbb{P}(h_B(x) = Y | X = x) - \mathbb{P}(h_Q(x) = Y | X = x)]$$

Now, we observe the point-wise relationship between regression gaps, which is true for all $x$:

$$\mathbb{P}(h_B(x) = Y | X = x) - \mathbb{P}(h_Q(x) = Y | X = x)$$
$$\leq L \cdot \mathbb{E}_{h \sim Q} \left( \mathbb{P}(h_B(x) = Y | X = x) - \mathbb{P}(Y = h(x) | X = x) \right)$$

Because of the majority rule nature of $h_Q$, if the output label $h_Q(x)$ is $l$, there must be at least $\frac{1}{L}$ (under $Q$) classifiers in $\mathcal{H}$ which predicts $l$. Hence, there must be at least $\frac{1}{L}$ (under $Q$) classifiers with the same gap in regression value as the voting classifier. In addition, because the other classifiers whose prediction is different from $h_Q(x)$ have non-negative regression gap, the statement is established. Integrating this point-wise inequality over $X$ and using monotonicity of integration, we are done. $\square$

In addition, the following lemma states that the distribution minimizing objectives that is the sum of a linear function in the distribution plus the Kullback-Leibler divergence w.r.t to a given priror has a particular exponential form.

**Lemma 2** (Lemma 1.1.3 of [17]). *Suppose $\mathcal{H}$ is a hypothesis class. Let $G : \mathcal{H} \to \mathbb{R}$ be a bounded function. For a reference distribution $P$, define the $Q^*$ distribution over $\mathcal{H}$:*

$$Q^*(h) = \frac{1}{c'} e^{-G(h)} P(h),$$

*where $c'$ is the normalization constant $c' = \mathbb{E}_{h \sim P} e^{-G(h)}$. Then, for all distributions Q over $\mathcal{H}$:*

$$\mathbb{E}_{h \sim Q} G(h) + \mathcal{D}_{kl}(Q \| P) = -\log \mathbb{E}_{h \sim P} e^{-G(h)} + \mathcal{D}_{kl}(Q \| Q^*).$$

We are ready to delve into the proof of Proposition 2.

**Proof of Proposition 2.** First, we show how the PAC-Bayes theorem of Proposition 1 applies to excess risk. It was necessary to introduce the intermediate loss function $e_h(x)$, which is valued in $[0, 1]$ while $e_h(x, y)$ might not be. The reason why $e_h(X_i)$ is valued in $[0, 1]$ is that:

$$e_h(x) = \mathbb{P}(h_B(X_i) = Y | X = x) - \mathbb{P}(h(x) = Y | X = x)$$

where $0 \leq \mathbb{P}(h(x) = Y | X = x) \leq \mathbb{P}(h_B(x) = Y | X = x) \leq 1$. In addition, $\mathcal{E}(h) = \mathbb{E} e_h(X)$. Therefore we can apply Proposition 1 with $l_h(x, y) = e_h(x)$ to conclude that, with probability at least $1 - \delta$ over the sampling of $\mathcal{S}$, simultaneously for all $\lambda \in (0, 2)$ and all posteriors $Q$:

$$\mathcal{E}(Q) \leq \frac{\widetilde{\mathcal{E}}(Q, \mathcal{S})}{1 - \lambda/2} + \frac{\mathcal{D}_{\text{kl}}(Q \| P) + \log \frac{2\sqrt{n}}{\delta}}{\lambda(1 - \lambda/2)n}. \tag{7}$$

Now, by the definition of $\widehat{\Delta}_n(h, \mathcal{S})$, with probability at least $1-\delta$ over the sampling of $\mathcal{S}$, $\forall \lambda \in (0, 2)$, for all $h \in \mathcal{H}$ it holds that $\widetilde{\mathcal{E}}(h, \mathcal{S}) \leq \widehat{\mathcal{E}}(h, \mathcal{S}) + \widehat{\Delta}_n(h, \mathcal{S})$. Because the upper bound holds for all classifiers, when we take expectation on both sides using the same distribution, the inequality still holds. In other words, for a distribution $Q$, if we define $\Delta_n(Q, \mathcal{S}) \doteq \mathbb{E}_{h \sim Q} \Delta_n(h, \mathcal{S})$ then with probability at least $1-\delta$ over $\mathcal{S}$, simultaneously for all $Q$:

$$\widetilde{\mathcal{E}}(Q, \mathcal{S}) \leq \widehat{\mathcal{E}}(Q, \mathcal{S}) + \widehat{\Delta}_n(Q, \mathcal{S}) \tag{8}$$

A simple union bound guarantees that the probability of both events in Equation 7 and Equation 8 occurring is at least $1 - 2\delta$. In such situations, simultaneously for all $\lambda \in (0, 2)$ and distributions $Q$:

$$\mathcal{E}(Q) \leq \frac{\widehat{\mathcal{E}}(Q, \mathcal{S}) + \widehat{\Delta}_n(Q, \mathcal{S})}{1 - \lambda/2} + \frac{\mathcal{D}_{\mathrm{kl}}(Q\|P) + \log \frac{2\sqrt{n}}{\delta}}{\lambda(1 - \lambda/2)n} \tag{9}$$

Second, we prove that for any fixed $\lambda \in (0, 2)$, $Q_\lambda^*$ as defined in Equation 2 minimizes the right hand side of Equation 9 out of all distributions over $\mathcal{H}$, which is equivalent to minimizing:

$$\phi(Q) \doteq \mathbb{E}_{h \sim Q}[n\lambda\widehat{\mathcal{E}}(h, \mathcal{S}) + n\lambda\widehat{\Delta}_n(h, \mathcal{S})] + \mathcal{D}_{\mathrm{kl}}(Q\|P). \tag{10}$$

This objective has the right form for us to apply Lemma 2. We define $G(h) = n\lambda\widehat{\mathcal{E}}(h, \mathcal{S}) + n\lambda\widehat{\Delta}_n(h, \mathcal{S})$. Because $\widehat{\Delta}_n(h, \mathcal{S})$ is bounded, $G(h)$ satisfies the boundedness assumption and we can apply the Lemma. Since the Kullback-Leibler divergence is non-negative, it is true that

$$\min_Q \phi(Q) = \min_Q (\mathbb{E}_{h \sim Q} G(h) + KL(Q\|P)) = -\log \mathbb{E}_{h \sim P} e^{-G(h)},$$

The minimum is attained at $Q = Q^*$. Upon closer inspection, it turns out that $Q^* = Q_\lambda^*$ as defined in Equation 2. Clearly, empirical excess risk is equal to the difference between empirical risks:

$$\widehat{\mathcal{E}}(h, \mathcal{S}) = \widehat{\mathcal{R}}(h, \mathcal{S}) - \widehat{\mathcal{R}}(h_B, \mathcal{S}).$$

Therefore:

$$Q^*(h) = \frac{e^{-n\lambda(\widehat{\mathcal{E}}(h, \mathcal{S}) + \widehat{\Delta}_n(h, \mathcal{S})} P(h)}{\mathbb{E}_{h' \sim P} e^{-n\lambda(\widehat{\mathcal{E}}(h, \mathcal{S}) + \widehat{\Delta}_n(h, \mathcal{S})}} = \frac{e^{n\lambda\widehat{\mathcal{R}}(h_B, \mathcal{S})} \cdot e^{-n\lambda(\widehat{\mathcal{R}}(h, \mathcal{S}) + \widehat{\Delta}_n(h, \mathcal{S})} P(h)}{e^{n\lambda\widehat{\mathcal{R}}(h_B, \mathcal{S})} \cdot \mathbb{E}_{h' \sim P} e^{-n\lambda(\widehat{\mathcal{R}}(h', \mathcal{S}) + \widehat{\Delta}_n(h', \mathcal{S}))}}$$

$$= \frac{e^{-n\lambda(\widehat{\mathcal{R}}(h, \mathcal{S}) + \widehat{\Delta}_n(h, \mathcal{S})} P(h)}{\mathbb{E}_{h' \sim P} e^{-n\lambda(\widehat{\mathcal{R}}(h', \mathcal{S}) + \widehat{\Delta}_n(h', \mathcal{S}))}} = Q_\lambda^*(h).$$

Third, we put the results of the first two steps to formulate an oracle inequality for the Gibbs classifier. For simplicity assume that some $h^* \in \mathcal{H}$ achieves the infimum in the right hand side of Proposition 2. By Hoeffding's inequality [18], since $\widehat{\mathcal{E}}(h^*, \mathcal{S}) = \frac{1}{n} \sum_{i=1}^n e_h(X_i, Y_i)$ where $e_h(X_i, Y_i)$ are i.i.d valued in $\{-1, 1\}$, it holds with probability at least $1 - \delta$ over $\mathcal{S}$ that:

$$\widehat{\mathcal{E}}(h^*, \mathcal{S}) - \mathcal{E}(h^*) \leq \sqrt{\frac{2\log \frac{1}{\delta}}{n}}. \tag{11}$$

In addition, also consider the following event, which has probability at least $1 - \delta$, where:

$$\widehat{\Delta}_n(h^*, \mathcal{S}) \leq \Delta_n(h^*) \tag{12}$$

Supposing that the statements of Equation 9, Equation 11 and Equation 12 hold, which is an event of probability at least $1 - 4\delta$. Because $Q_\lambda^*$ minimizes the right hand side of Equation 9, in particular it does better than $Q_{h^*}$ i.e.

$$\mathcal{E}(Q_\lambda^*) \leq \frac{n\lambda(\widehat{\mathcal{E}}(h^*, \mathcal{S}) + \widehat{\Delta}_n(h^*, \mathcal{S})) - \log P(h^*) + \log \frac{2\sqrt{n}}{\delta}}{\lambda(1 - \frac{\lambda}{2})n}$$

$$\leq \frac{n\lambda(\mathcal{E}(h^*) + \sqrt{2\frac{\log \frac{1}{\delta}}{n}} + \Delta_n(h^*)) - \log P(h^*) + \log \frac{2\sqrt{n}}{\delta}}{\lambda(1 - \frac{\lambda}{2})n}$$

$$\leq \frac{\mathcal{E}(h^*) + \Delta_n(h^*)}{1 - \frac{\lambda}{2}} + \frac{\log(1/P(h^*)) + \log \frac{2\sqrt{n}}{\delta} + \lambda\sqrt{2n\log \frac{1}{\delta}}}{\lambda(1 - \frac{\lambda}{2})n}.$$

Finally, combining this upper bound on the excess risk of the Gibbs classifier with Lemma 1 to conclude that with probability at least $1 - 4\delta$ over $\mathcal{S}$:

$$\mathcal{E}(h_{Q^*_\lambda}) \leq \frac{L}{1 - \lambda/2} \inf_{h \in \mathcal{H}} \left( \mathcal{E}(h) + \Delta_n(h) + \frac{\log(1/P(h))}{\lambda n} + \frac{\log \frac{2\sqrt{n}}{\delta} + \lambda \sqrt{2n \log \frac{1}{\delta}}}{\lambda n} \right).$$

$\square$

## 6.2 Theorem 1

We need a few lemmas to prove Theorem 1. The first is Kraft's inequality, a standard tool in coding theory.

**Lemma 3** (Kraft's inequality [19]). *For any prefix-free code over an alphabet of size D, the codeword lengths $l_1, l_2, \ldots, l_m$ must satisfy:*

$$\sum_i D^{-l_i} \leq 1$$

The next lemma shows that the normalization constant defining the prior in Theorem 1 is at most 1.

**Lemma 4.** *Recall that the prior in Theorem 1 has the normalization constant $C_P = \sum_{h_T \in \mathcal{H}(T_0)} e^{-3D_0 \cdot |\pi(T)|}$. It is true that:*

$$C_P \leq 1$$

*Proof.* The idea is to design a prefix-free codebook for $\mathcal{H}(T_0)$ over the binary alphabet $\{0, 1\}$ and use Kraft's inequality, a standard tool in coding theory (A prefix-free code is a codebook such that no codeword is a prefix of another). Because of the one-to-one correspondence between the classifier $h_T$ and the subtree $T$, it suffices to encode $T$.

First, we define a prefix-free code for the set of nodes in $T_0$ by modifying the strategy in Section 2.3.2 of [16]. The code for a node $A$ consists of three components, concatenated in order of appearance:

- Depth encoding: if the depth of $A$ is $k$, the encoding is string of $k$ ones.

- A delimiting 0 to signify that the depth encoding has ended

- Path encoding: the sequence of left and right links that is the path from the root to $A$, where a left link is encode as 0 and a right link is encoded as 1.

This is a prefix-free code for the set of nodes of $T_0$: the sequence of ones at the start eliminates the possibilities of two nodes at different depths having codewords that are prefix of each other, and for nodes of the same depth, there will be a discrepancy in the paths from root to the nodes that makes it impossible for prefixing. Given this codebook of nodes, the encoding $E$ of subtree $T$, it suffices to concatenate the codewords for $A \in \pi(T)$: deeper nodes are put in front, and among those at the same depth, go from left to right.

Second, we prove that $E$ is a prefix-free code for the family of subtrees $T$ over the alphabet $\{0, 1\}$ i.e. for subtrees $T_1 \neq T_2$, neither $E(T_1)$ is a prefix of $E(T_2)$ nor vice versa. On one hand, consider the case that $E(T_1) = E(T_2)$ i.e. two different subtrees having the same encoding. However, since no two different subtrees can have the same leaf set, this is not possible.

On the other hand, consider the case when one codeword is a proper prefix of the other: without loss of generality, assume $E(T_1)$ is a proper prefix of $E(T_2)$. Because $E$ is the concatenation of prefix-free codes, from $E(T_1)$ we reconstruct uniquely the leaf set $\pi(T_1)$ and from $E(T_2)$ we reconstruct uniquely the leaf set $\pi(T_2)$. Since $E(T_1)$ is a proper prefix of $E(T_2)$, it must be true that one leaf set is contained in the other $\pi(T_1) \subset \pi(T_2)$ and there exists $A'$ such that $A' \in \pi(T_2)$ but $A' \notin \pi(T_1)$. But because both $\pi(T_1)$ and $\pi(T_2)$ partition $\mathcal{X}$, it means that $A' \cap \mathcal{X} = \emptyset$. This is a contradiction since $A'$ is supposed to intersect $\mathcal{X}$

Overall, there can be no code that is a prefix of another code. We now analyze the length of the encoding $E$. It is easy to see that each node $A$ has a codelength at most 3 times its depth. Therefore, to encode a subtree $T$, whose maximum depth is bounded by $D_0$ and has $|\pi(T)|$ leaves we can

use codewords whose lengths are upper bounded by $3 \log_2 e \cdot D_0 \cdot |\pi(T)|$. We now employ Kraft's inequality. In our case, the prefix-free codebook for subtrees $T$ is over the alphabet $\{0, 1\}$ and $3D_0 \log_2 e \cdot |\pi(T)|$ are upper bounds on the codelengths. Hence:

$$\sum_{h_T \in \mathcal{H}(T_0)} 2^{-(3D_0 \log_2 e \cdot |\pi(T)|)} \leq 1 \implies \sum_{h_T \in \mathcal{H}(T_0)} e^{-3D_0 \cdot |\pi(T)|} \leq 1.$$

$\square$

The next lemma proves that $\widehat{\Delta}_n(h_T, \mathcal{S})$ and $\Delta_n(h_T)$ defined in Equation 3 and Equation 4 satisfies the conditions of Proposition 2 for the hypothesis class $\mathcal{H}(T_0)$.

**Lemma 5.** *With $\widehat{\Delta}_n(h_T, \mathcal{S})$ defined as in Equation 3, with probability at least $1 - \delta$ over $\mathcal{S}$, for all $h_T \in \mathcal{H}(T_0)$:*

$$\widetilde{\mathcal{E}}(h_T, \mathcal{S}) \leq \widehat{\mathcal{E}}(h_T, \mathcal{S}) + \widehat{\Delta}_n(h_T, \mathcal{S})$$

*Proof.* Let $n(A, \mathcal{S}) = n\hat{p}(A, \mathcal{S})$ be the number of data points in $\mathcal{S}$ which falls into $A$. Define

$$c(A, \mathcal{S}) \doteq \begin{cases} \frac{1}{n(A,\mathcal{S})} \sum_{X_i \in A} e_h(X_i, Y_i) & \text{if } n(A, \mathcal{S}) > 0 \\ 0 & \text{otherwise} \end{cases}$$

For fixed $X^n = \{X_i\}_{i=1}^n$, denote by $\bar{c}(A, X^n)$ the condition expectation over $Y^n = \{Y_i\}_{i=1}^n$ is:

$$\bar{c}(A, X^n) \doteq \mathbb{E}_{Y^n|X^n} c(A, \mathcal{S})$$
$$= \begin{cases} \frac{1}{n(A,\mathcal{S})} \sum_{X_i \in A} e_h(X_i) & \text{if } n(A, \mathcal{S}) > 0 \\ 0 & \text{otherwise} \end{cases}$$

First, for fixed $X^n$, we derive a uniform concentration result for all nodes $A$ in $T_0$, with the randomness from $Y^n$. For nodes that contain data, $c(A, \mathcal{S})$ is the average of $n(A, \mathcal{S})$ independent random variables since $Y_1, Y_2, \ldots, Y_n$ are independent conditioned on $X^n$. Furthermore, $c(A, \mathcal{S})$ satisfies a bounded variation condition: a change in some $Y_i$ for results in at most a change of $\frac{2}{n(A,\mathcal{S})}$ in $c(A, \mathcal{S})$. Hence, for any node $A$ of $T_0$, for any $\epsilon_A > 0$, by McDiarmid's inequality [20]:

$$\Pr_{Y^n|X^n} (\bar{c}(A, X^n) - c(A, \mathcal{S}) > \epsilon_A) \leq \exp\left( -\frac{2\epsilon_A^2}{n(A, \mathcal{S})(\frac{2}{n(A,\mathcal{S})})^2} \right) = e^{-n(A,\mathcal{S})\epsilon_A^2/2}$$

We set $\epsilon_A = \sqrt{2 \log(|T_0| / \delta)/n(A, \mathcal{S})}$. In addition, we multiply both sides of the event $\bar{c}(A, X^n) - c(A, \mathcal{S}) > \epsilon_A$ by $\hat{p}(A, \mathcal{S})$ to have that:

$$\Pr_{Y^n|X^n} \left( \hat{p}(A, \mathcal{S})(\bar{c}(A, X^n) - c(A, \mathcal{S})) \leq \sqrt{\hat{p}(A, \mathcal{S})\frac{2 \log(|T_0| / \delta)}{n}} \right) \leq \frac{\delta}{|T_0|}$$

Now, we perform a union bound, with at most $|T_0|$ elements to conclude that for fixed $X^n$, with probability at least $1 - \delta$ over $Y^n$, simultaneously for all $A$ such that $\hat{p}(A, \mathcal{S}) > 0$:

$$\hat{p}(A, \mathcal{S})(\bar{c}(A, X^n) - c(A, \mathcal{S})) \leq \sqrt{\hat{p}(A, \mathcal{S})\frac{2 \log(|T_0| / \delta)}{n}} \tag{13}$$

As for nodes $A$ such that $\hat{p}(A, \mathcal{S}) = 0$, by definition, for all $Y^n$ it holds that that $\bar{c}(A, X^n) - c(A, \mathcal{S}) = 0$. Hence, the statement in Equation 13 is also true for empty cells. Moving on to the tail bound for the subtree classifiers' excess risk. The empirical excess risk, the intermediate losss and the penalty of each $h_T$ is decomposable over the leaves of $T$:

$$\widetilde{\mathcal{E}}(h_T, \mathcal{S}) - \widehat{\mathcal{E}}(h_T, \mathcal{S}) = \sum_{A \in \pi(T)} \hat{p}(A, \mathcal{S})[\bar{c}(A, X^n) - c(A, \mathcal{S})]$$

$$\widehat{\Delta}_n(h_T, \mathcal{S}) = \sum_{A \in \pi(T)} \sqrt{\hat{p}(A, \mathcal{S})\frac{2 \log(|T_0| / \delta)}{n}}$$

Because of this decomposition, the following inclusion of events is true:

$$\{\exists h_T : \widetilde{\mathcal{E}}(h_T, \mathcal{S}) - \widehat{\mathcal{E}}(h_T, \mathcal{S}) > \widehat{\Delta}_n(h_T, \mathcal{S})\}$$

$$\subseteq \left\{ \exists A : \hat{p}(A, \mathcal{S})[\bar{c}(A, X^n) - c(A, \mathcal{S})] > \sqrt{\hat{p}(A, \mathcal{S}) \frac{2 \log(|T_0|/\delta)}{n}} \right\}$$

According to Equation 13, the probability over $Y^n$ of the later event is at most $\delta$. Therefore:

$$\Pr_{Y^n|X^n} (\exists h_T : \widetilde{\mathcal{E}}(h_T, \mathcal{S}) - \widehat{\mathcal{E}}(h_T, \mathcal{S}) > \widehat{\Delta}_n(h_T, \mathcal{S})) \leq \delta$$

Taking the expectation of both sides w.r.t to $X^n$, and taking the complement event, we conclude that:

$$\Pr_{\mathcal{S}}(\forall h_T : \widetilde{\mathcal{E}}(h_T, \mathcal{S}) - \widehat{\mathcal{E}}(h_T, \mathcal{S}) \leq \widehat{\Delta}_n(h_T, \mathcal{S})) \geq 1 - \delta$$

$\square$

**Lemma 6.** *With $\Delta_n(h_T)$ defined in Equation 4 for any $h_T \in \mathcal{H}(T_0)$, with probability at least $1 - \delta$ over $\mathcal{S}$:*

$$\widehat{\Delta}_n(h_T, \mathcal{S}) \leq \Delta_n(h_T)$$

*Proof.* It is implied by Lemma 1 [16]. Each node in $T_0$ can be associated with a codeword $\|A\|$ that is proportional to its depth in $T_0$, with the constant being upper bounded by $2 + \log D$. Then, with probability at least $1 - \delta$, for all $A$:

$$\hat{p}(A, \mathcal{S}) \leq 4 \max \left( p(A), \frac{\|A\| + \log(1/\delta)}{n} \right)$$

Since the maximal depth of a node in $T_0$ is $D_0$, we replace $\|A\|$ by $(2 + \log D) \cdot D_0$:

$$\hat{p}(A, \mathcal{S}) \leq 4 \max \left( p(A), \frac{(2 + \log D) \cdot D_0 + \log(1/\delta)}{n} \right)$$

If the inequality above holds for all $A$, for any $h_T$, by taking the summation of $A \in \pi(T)$ on both sides to prove the statement of the lemma. $\square$

**Proof of Theorem 1.** Theorem 1 is a direct application of Proposition 2 for the family $\mathcal{H}(T_0)$.

It is clear from Lemma 5 and Lemma 6 that $\widehat{\Delta}_n(h_T, \mathcal{S})$ and $\Delta_n(h_T)$ satisfy the conditions of Proposition 2 (the boundedness condition is automatically satisfied because $\mathcal{H}(T_0)$ is finite). Hence, for the choice of prior $P(h_T) = \frac{1}{C_p} e^{-3D_0 \cdot |\pi(T)|}$, with probability at least $1 - 4\delta$ over $\mathcal{S}$, simultaneously for all $\lambda \in (0, 2)$:

$$\mathcal{E}(h_{Q_\lambda^*})$$

$$\leq \frac{L}{1 - \lambda/2} \min_{h_T \in \mathcal{H}(T_0)} \left( \mathcal{E}(h_T) + \Delta_n(h_T) + \frac{\log C_p + 3D_0 \cdot |\pi(T)|}{\lambda n} + \frac{\log \frac{2\sqrt{n}}{\delta} + \lambda \sqrt{2n \log \frac{1}{\delta}}}{\lambda n} \right).$$

By Lemma 4, $\log C_P \leq 0$. This shows the statement of the Theorem. $\square$

## 6.3 Corollary 1

As usual when it comes to tree-based classification, we first go through tree-based regression. Each label $Y$ is converted to a vector one-hot encoding $b(Y)$ where the $l$ coordinate $b_l(Y) = 1$ if $Y = l$ and 0 otherwise. Then we have a family of regression trees $\{\eta_T\}$ indexed by subtrees $T$ of $T_0$ where $\eta_T$ defines a fixed regression value $s(A)$ of nodes $A \in \pi(T)$, e.g. $s(A) \doteq$ average value of $b(Y)$ if $A \cap \mathcal{S}_0 \neq \emptyset$.

The following proposition, implied by Equation A.1 in [13], shows the bias-variance decomposition of $L_2$ excess risk for regressors based on dyadic trees.

**Proposition 3** (Bias-variance decomposition [13]). *There are absolute constants $C_3, C_4$ such that the following hold. For any $T$ that induces a dyadic partition of $\mathcal{X}$, it is true that:*

$$\mathbb{E}_{\mathcal{S}_0, X} \|\eta_T(X) - \eta(X)\|^2 \leq C_3 \lambda^2 r(T)^{2\alpha} + C_4 \frac{|\pi(T)|}{n}$$

The next lemma shows how the $L_2$ excess risk of a regressor is an upper bound on the excess risk of the associated plug-in classifier. The result for binary classification is well-established: here we prove in the multiclass case for completeness.

**Lemma 7.** *Suppose $\hat{\eta}(x)$ is a regressor (trained from data) and $\hat{h}(x)$ is the associated plug-in classifier $\hat{h}(x) = \operatorname{argmax}_{l \in [L]} \hat{\eta}_l(x)$. Then:*

$$\mathcal{E}(\hat{h}) \leq 2\sqrt{L} \cdot \sqrt{\mathbb{E}_X \|\hat{\eta}(X) - \eta(X)\|_2^2}$$

*Proof.* We first prove that:

$$\mathcal{E}(\hat{h}) \leq 2\mathbb{E}_X \|\hat{\eta}(X) - \eta(X)\|_1$$

The idea is to prove the point-wise inequality:

$$\mathbb{P}\left(h_B(x) = Y | X = x\right) - \mathbb{P}\left(\hat{h}(x) = Y | X = x\right) \leq 2|\hat{\eta}(x) - \eta(x)|$$

(Integration over $x$ of the left hand side gives the excess risk of $\hat{h}$ while integrating over the right hand side gives the $L_1$ regression risk.) The inequality can be rewritten as:

$$\eta_{h_B(x)}(x) - \eta_{\hat{h}(x)}(x) \leq 2|\hat{\eta}(x) - \eta(x)|$$

Let $|\hat{\eta}(x) - \eta(x)| = u$. Then, for all $l \in [L]$, $|\hat{\eta}_l(x) - \eta_l(x)| \leq u$. In particular:

$$\left|\hat{\eta}_{h_B(x)}(x) - \eta_{h_B(x)}(x)\right| \leq u$$

$$\left|\hat{\eta}_{\hat{h}(x)}(x) - \eta_{\hat{h}(x)}(x)\right| \leq u$$

From the first equation, we have $\eta_{h_B(x)}(x) \leq \hat{\eta}_{h_B(x)}(x) + u$, so that $\eta_{h_B(x)}(x) - \eta_{\hat{h}(x)}(x) \leq \hat{\eta}_{h_B(x)}(x) - \eta_{\hat{h}(x)}(x) + u$. Because $\hat{h}(x)$ is the plug-in of $\hat{\eta}(x)$, we have $\hat{\eta}_{h_B(x)}(x) \leq \hat{\eta}_{\hat{h}(x)}(x)$. Combined this fact with the second equation, which says $\hat{\eta}_{\hat{h}(x)}(x) - \eta_{\hat{h}(x)}(x) \leq u$, overall we have shown that $\eta_{h_B(x)}(x) - \eta_{\hat{h}(x)}(x) \leq 2u$.

We then combine with the well-known inequality between $L_p$ norms:

$$\mathbb{E}_X \|\hat{\eta}(X) - \eta(X)\|_1 \leq \sqrt{L \cdot \mathbb{E}_X \|\hat{\eta}(X) - \eta(X)\|_2^2}$$

Truly:

$$\|\hat{\eta}(X) - \eta(X)\|_1^2 = \left(\sum_l |\hat{\eta}_l(X) - \eta_l(X)|\right)^2 \leq L \sum_l (\hat{\eta}_l(X) - \eta_l(X))^2$$

so we have, by Jensen's inequality:

$$(\mathbb{E}_X \|\hat{\eta}(X) - \eta(X)\|)^2 \leq \mathbb{E}_X \|\hat{\eta}(X) - \eta(X)\|_1^2 \leq L \cdot \mathbb{E}\|\hat{\eta}(X) - \eta(X)\|_2^2$$

$\square$

**Proof of Corollary 1.** We first convert Theorem 1, a statement in high probability over $\mathcal{S}$, to a statement in expectation over $\mathcal{S}$. We select $\delta = \frac{1}{n}$. In the rare event (probability at most $\frac{4}{n}$) that the upper bound in Theorem 1 does not hold, we still have the trivial upper bound on excess risk $\mathcal{E}(h_{Q_\lambda^*}) \leq 1$. Therefore, for fixed $\mathcal{S}_0$ but taking expectation over $\mathcal{S}$, we have:

$$\mathbb{E}_{\mathcal{S}} \mathcal{E}(h_{Q_\lambda^*}) \leq C_0 \min_{h_T \in \mathcal{H}(T_0)} \left(\mathcal{E}(h_T) + \Delta_n(h_T) + \frac{3D_0 |\pi(T)|}{\lambda n} + \frac{\log(2n\sqrt{n}) + \lambda\sqrt{2n\log n} + 4\lambda}{\lambda n}\right).$$
(14)

where $C_0 = \frac{L}{1-\lambda/2}$. We now show that the expectation of the right hand side has the right dependencies on $n, \alpha, d$. The strategy is to demonstrate the existence of a right resolution $r$ that results in a classification tree with the right excess risk. By Assumption 2, for any $(C_1/n) < r \leq 1$, there exists a subtree $T^r$ of $T_0$ such that $r(T^r) \leq r$, $|\pi(T^r)| \leq C_2 r^{-d}$. Combined with Proposition 3, the excess risk of the regressor has the form:

$$\mathbb{E}_{\mathcal{S}_0,X}\|\eta_{T^r}(X) - \eta(X)\|^2 \leq C_3\lambda^2 r^{2\alpha} + C_4 \frac{r^{-d}}{n}.$$

Consider the choice $r_* \doteq (\frac{C_4}{C_3\lambda^2})^{1/(2\alpha+d)}(\frac{\log n}{n})^{1/(2\alpha+d)}$. Such $r_*$ is permissible since $\frac{1}{n}$ is smaller that $(\frac{\log n}{n})^{1/(2\alpha+d)}$ for all large $n$. Therefore, for some constant $C_5$, the $L_2$ excess risk of $\eta_{T^{r_*}}$ satisfies:

$$\mathbb{E}_{\mathcal{S}_0,X}\|\eta_{T^{r_*}}(X) - \eta(X)\|^2 \leq C_5\left(\frac{\log n}{n}\right)^{2\alpha/(2\alpha+d)}.$$

Because of how we define $\eta_T$, $h_T$ is the plug-in classifier associated with $\eta_T$. Using Lemma 7, we have for some constant $C_8$:

$$\mathbb{E}_{\mathcal{S}_0}\mathcal{E}(h_{T^{r_*}}) \leq C_8\left(\frac{\log n}{n}\right)^{\alpha/(2\alpha+d)}.$$

We move on to bound $\Delta_n(h_{T^{r_*}})$. Observe that $\sqrt{\max(a,b)} \leq \sqrt{a} + \sqrt{b}$ for any $a, b \geq 0$. Hence:

$$\Delta_n(h_{T^{r_*}}) = \sqrt{\frac{\log(n\,|T_0|)}{n}} \sum_{A\in\pi(T^{r_*})} \sqrt{4\max\left(p(A), \frac{D_0 + \log n}{n}\right)}$$

$$\leq 2\sqrt{\frac{\log(n\,|T_0|)}{n}} \sum_{A\in\pi(T^{r_*})} \left(\sqrt{p(A)} + \sqrt{\frac{D_0 + \log n}{n}}\right)$$

$$\leq 2\sqrt{\frac{\log(n\,|T_0|)}{n}} \sum_{A\in\pi(T^{r_*})} \sqrt{p(A)} + 2\sqrt{\log(n\,|T_0|)(D_0 + \log n)}\frac{|\pi(T^{r_*})|}{n}.$$

We bound the first summation by Jensen's inequality for concave $\sqrt{x}$, supposing that the summation is over $A$ such that $p(A) > 0$:

$$\sum_{A\in\pi(T^{r_*})} \sqrt{p(A)} = \sum_{A\in\pi(T^{r_*})} p(A)\frac{1}{\sqrt{p(A)}} \leq \sqrt{\sum_{A\in\pi(T^{r_*})} \frac{p(A)}{p(A)}} = \sqrt{|\pi(T^{r_*})|}.$$

Combining the fact that the maximal depth $D_0 = O(D\log n)$ and that $T_0$ has $O(n)$ leaves, it means that $|T_0| = O(Dn\log n)$. Therefore, there exists a constant $C_6$ such that:

$$\Delta_n(h_{T^{r_*}}) \leq C_6(\sqrt{\frac{\log n \cdot |\pi(T^{r_*})|}{n}} + \frac{\log n \cdot |\pi(T^{r_*})|}{n}).$$

Recall that $|\pi(T^{r_*})| \leq r_*^{-d}$. This implies that for setting $r = r^*$ there exists some constant $C_7$ such that

$$\sqrt{\frac{\log n \cdot |\pi(T^{r_*})|}{n}} \leq C_7\left(\frac{\log n}{n}\right)^{\alpha/(2\alpha+d)},$$

which leads to the overall conclusion that there exists some constant $C_8$ satisfying:

$$\mathbb{E}_{\mathcal{S}_0}\left(\mathcal{E}(h_{T^{r_*}}) + \Delta_n(h_{T^{r_*}}) + \frac{3D_0 \cdot |\pi(T^{r_*})|}{\lambda n}\right) \leq C_8\left(\frac{\log n}{n}\right)^{\alpha/(2\alpha+d)}. \tag{15}$$

The order of growth on the right hand side of the above inequality dominates the $\frac{\log(2n\sqrt{n})+\lambda\sqrt{2n\log n}+4\lambda}{\lambda n}$ component of the right hand side of Equation 14. We now show the rate of convergence over $\mathcal{S}_0$ and $\mathcal{S}$. Clearly, for any $\mathcal{S}_0$:

$$\min_{h_T\in\mathcal{H}(T_0)} \left(\mathcal{E}(h_T) + \Delta_n(h_T) + \frac{3D_0 \cdot |\pi(T)|}{\lambda n}\right) \leq \mathcal{E}(h_{T^{r_*}}) + \Delta_n(h_{T^{r_*}}) + \frac{3D_0 \cdot |\pi(T^{r_*})|}{\lambda n}. \tag{16}$$

Therefore, by taking expectation of both sides of Equation 14, combining Equations 15 and 16, we conclude that there exists a constant $C$ such that:

$$\mathbb{E}_{\mathcal{S}_0,\mathcal{S}}\mathcal{E}(h_{Q_\lambda^*}) \leq C \left(\frac{\log n}{n}\right)^{\alpha/(2\alpha+d)}.$$

$\square$

## 6.4 Theorem 2

*Proof.* That the combined runtime of Algorithm 1 and Algorithm 2 is $2\left|\bar{T}_0\right|$ is clear: and each algorithm will visit each node of $\bar{T}_0$ exactly once, doing a constant amount of computation. In addition, $\left|\bar{T}_0\right| \leq 2\left|T_0\right|$ since we only add one dummy node to at most $|T_0|$ nodes.

Regarding the correctness of the procedure: if we show that indeed, the $\beta(A)$ computed through Algorithm 1 is equal to the right hand side of Equation 6 and the $\alpha(A)$ computed through Algorithm 2 is equal to the right hand side of Equation 5, then we will have proven that after the two algorithms, $w(A) = Cw_{Q_\lambda^*}(A)$ for some positive constant $C$, since if $\alpha(A)$ is the expression in Equation 5, then

$$w(A) = e^{\phi(A)} \sum_{h_T:A\in\pi(T)} \exp\left(\sum_{A'\neq A, A'\in\pi(T)} \phi(A')\right) = \sum_{h_T:A\in\pi(T)} \exp\left(\sum_{A'\in\pi(T)} \phi(A')\right)$$
$$= C_{Q_\lambda^*} \cdot w_{Q_\lambda^*}(A)$$

The strategy is to prove the correctness on $\bar{T}_0$: because the dummy nodes $A'$ that we add to $T_0$ to form $\bar{T}_0$ have zero contribution $\phi(A') = 0$, we also have correctness on $T_0$. For $A$ in $\bar{T}_0$, we denote by

$$\beta^*(A) \doteq \sum_{T\preceq\bar{T}_0^A} \exp\left(\sum_{A'\in\pi(T)} \phi(A')\right),$$

and

$$\alpha^*(A) \doteq \sum_{T:A\in\pi(T)} \exp\left(\sum_{A'\neq A, A'\in\pi(T)} \phi(A')\right).$$

and set out to prove that $\beta(A) = \beta^*(A)$ and $\alpha(A) = \alpha^*(A)$. For any node $A$, denote $A_L$ to be the left child of $A$ and $A_R$ to be the right child, in the augmented tree $\bar{T}_0$.

Regarding $\beta(A) = \beta^*(A)$, it suffices to show that $\beta^*(A)$ satisfies the base case and the recurrence relation defining $\beta(A)$ in Algorithm 1, namely:

$$\beta^*(A) = \begin{cases} e^{\phi(A)} & \text{if } A \in \pi(\bar{T}_0) \\ e^{\phi(A)} + \beta^*(A_L)\beta^*(A_R) & \text{otherwise} \end{cases}$$

When $A$ is a leaf, $\beta^*(A) = e^{\phi(A)}$ is immediate since the only the subtree rooted at $A$ is $A$ itself, viewed as a subtree. When $A$ is an internal node, any $T \preceq T_0^A$ is either just the node $A$ or can be decomposed into a left subtree $T_L$ and a right subtree $T_R$ that are rooted at $A_L$ and $A_R$, respectively. The former case contributes the term $e^{\phi(A)}$ to the sum. In the later case, the product over $A' \in \pi(T)$ is the same as the product over $A_1 \in \pi(T_L)$ times the product over $A_2 \in \pi(T_R)$. In other words:

$$\beta^*(A) = e^{\phi(A)} + \sum_{T_L\preceq T_0^{A_L}} \sum_{T_R\preceq T_0^{A_R}} \prod_{A_1\in\pi(T_L)} \prod_{A_2\in\pi(T_R)} e^{\phi(A_1)}e^{\phi(A_2)}$$
$$= e^{\phi(A)} + \left(\sum_{T_L\preceq T_0^{A_L}} \prod_{A_1\in\pi(T_L)} e^{\phi(A_1)}\right)\left(\sum_{T_R\preceq T_0^{A_R}} \prod_{A_2\in\pi(T_R)} e^{\phi(A_2)}\right)$$
$$= e^{\phi(A)} + \beta^*(A_L)\beta^*(A_R)$$

Hence, we have $\beta(A) = \beta^*(A)$.

Regarding $\alpha(A) = \alpha^*(A)$, again we aim to show that $\alpha^*(A)$ satisfies the base case and recurrence relation defining $\alpha(A)$ in Algorithm 2, namely:

$$\alpha^*(\text{root}) = 1$$
$$\alpha^*(A_L) = \alpha^*(A)\beta^*(A_R)$$
$$\alpha^*(A_R) = \alpha^*(A)\beta^*(A_L)$$

The first equation is clear: when $A$ is the root node, the summation defining $\alpha^*(A)$ is over the subtree with only the root, and there are no leaves in this tree except the root, so the summation inside the exponential is $0$. We only need to prove the second equation: the third follows in the same manner. The partition $T$ of any subtree classifier $h_T$ such that $A_L \in \pi(T)$ can be decomposed into three parts: the part $T_1$ which is a pruned subtree of $T_0$ such that $A \in \pi(T_1)$, the extension of $T_1$ by a pruned subtree rooted at $A_R$ (which we denote $T_2$) and the inclusion of itself $A_L$ as a leaf. There are no constraints between $T_1$ and $T_2$ except that the former must have $A$ as a leaf and $T_2$ is rooted at $A_R$. Therefore:

$$\alpha^*(A_L) = \sum_{T_1 : A \in \pi(T_1)} \sum_{T_2 \preceq T_0^{A_R}} \exp\left( \sum_{A' \neq A, A' \in \pi(T_1)} \phi(A') \right) \exp\left( \sum_{A' \in \pi(T_2)} \phi(A') \right)$$

$$= \left( \sum_{T_1 : A \in \pi(T_1)} \exp\left( \sum_{A' \neq A, A' \in \pi(T_1)} \phi(A') \right) \right) \left( \sum_{T_2 \preceq T_0^{A_R}} \exp\left( \sum_{A' \in \pi(T_2)} \phi(A') \right) \right)$$

$$= \alpha^*(A)\beta^*(A_R)$$

Hence we have $\alpha(A) = \alpha^*(A)$. $\qquad\square$