[Reviews · NeurIPS 2018]

Reviewer 1



**Summary and main remarks** The manuscript investigates generalization bounds for classification trees, using a PAC-Bayesian approach. Such generalization bounds exist in the literature for two competing strategies to build classification trees: given a fully developed tree, (i) pruning and (ii) weighted majority vote on all subtrees. Pruning is by far the most popular strategy, since it enjoys computational and statistical guarantees. The main claim from the manuscript is that the authors devise a weighted majority vote strategy which has similar generalization bounds and a computational complexity bounded by the total nomber of nodes of the whole tree. The workhorse of this approach is the PAC-Bayesian theory, which allows to derive oracle generalization bounds for randomized classifiers sampled from a Gibbs potential. Up to my knowledge, the use of PAC-Bayesian tools to analyze classification trees is original. Existing PAC-Bayesian bounds hold for randomized classifiers sampled from the Gibbs potential, and the passing to a Q-weighted majority vote (where Q is a distribution on the set of classifiers -- in the manuscript, this set is assumed finite) is not trivial, and is identified as an issue in earlier papers (e.g., Lacasse, Laviolette, Marchand, Germain and Usunier, 2007). The authors adapt an existing PAC-Bayesian bound from Thiemann, Igel, Wintenberger and Seldin (beware, the reference details are wrong), and the cornerstone of their analysis is stated in Proposition 2 (page 4). The crux is to twist the loss function to take into account the different nature of the classifier (randomized vs. aggregate / expectation). Proposition 2 is then a straightforward consequence of the previous bound from Thiemann et al. This is a simple yet elegant argument. The authors then adapt their version of a PAC-Bayesian bound for Q-weighted majority vote to the case of trees and aggregates of subtrees: Theorem 1 (page 5). This oracle bound which is presented states that with arbitrarily high probability, the excess risk of the Q-weighted majority vote is upper bounded by the best (= smallest) excess risk of the subtrees, up to remainder terms of magnitude $\mathcal{O}(D_0 |\pi(T)|/(n\lambda) + \sqrt{n})$ *and* up to a multiplicative constant which is $L/(1-\lambda/2)$. This Theorem 1 bears my main concerns about the manuscript. (i) why not optimize with respect to $\lambda$? I agree with the comment on lines 129--131 yet the bound would be sharper and easier to read. I am a bit confused about the fact that $\lambda$ is not taken care of in the theory even though its optimization seems straightforward, and automatically selected via cross-validation in the numerical experiments. Have I missed something there? (ii) the leading constant is never 1 as soon as $L\geq 2$, thus the oracle inequality is not sharp. This clearly undermines the conclusion as the excess risk is not comparable to the risk of the oracle (the best subtree, in the sense of the smallest excess risk). I reckon that obtaining a sharp inequality might not be possible with the scheme of proofs used by the authors yet I would find interesting to see comments on this. (iii) at first glance, the rate seemed to be 1/n but it is actually 1/\sqrt{n}, which is much more coherent given the assumptions made in the manuscript. I feel the authors should more clearly state this fact, to avoid ambiguity and possible confusion. The manuscript then moves on to the (favorable) situation where the excess risk of the oracle is known to converge to 0. This is done at the price of a contraction bound on the regression functions $\mathbb{E \mathbf{1}_{Y=\ell}$ with an exponent parameter $\alpha$. I am a bit puzzled by this assumption as the regression functions take their values in $(1,L)$ and that the diameter of the input space is 1. So I feel that this assumption might be quite stringent and implies a rather strong constraint on the unknown underlying distribution of $(X,Y)$. An additional comment might be useful there. In this setting, the rate of convergence attained by the oracle is the classical nonparametric rate $n^{-\alpha/(2\alpha+d)}$ where $d$ stands for the intrinsic dimension rather than the dimension $D$ of the input space. While very classical in nonparametric statistics, I fear that $d$ is, in practice, quite large for most classification trees, and the rate becomes arbitrarily slow. Nevertheless, this remark is a bit beyond the scope of the manuscript since the key message here is that whenever the oracle is consistent (its excess risk converges to 0), the Q-weighted majority vote inherits the rate of convergence of the oracle. The rate does not really matter to draw that conclusion. This is a very classical conclusion in aggregation literature yet it is the first time I see this for classification trees. **Overall assessment** The manuscript is well-written and carries interesting results, using ideas which are not novel but never combined so far. The scope of the work is of interest to the NIPS community. However I have some concerns about the main results. **Specific remarks** - In the references: improper capitalization of words such as PAC, Bayes / Bayesian. Reference 14 is incorrect (remove "et al."). Reference 9 is incorrect (it has been long published, at ALT 2017). - LaTeX tip: in math mode, use \ell instead of l, this us much more pleasant to the eye! [Rebuttal acknowledged. I am satisfied with the answers provided by the authors and I upgrade my score to 7.]

Reviewer 2



This is a paper on the derivation and use of PAC-Bayesian results for the problem of learning classification trees. It provides - theoretical results for the model that consists in weighting the subtrees of a general tree to have an efficient (multiclass) classifier; - algorithmic strategy to efficiently compute the weighting of each subtree from weights associated with all the nodes of the "base" tree; - empirical evidence that the theory-guided learning procedure is indeed relevant. The paper is extremely clear and easy to follow. Going from theoretical results to algorithmic procedures to compelling results is something that is very important in machine learning. This paper combines these three features. I just have some questions/remarks: - is it possible to provide the value of the bound on the excess risk? that would indicate how precise the Pac-Bayesian results are. - Germain et al, worked a lot on the connection between PAC-Bayesian bounds, the Gibbs risk and the possibility to use the majority vote, together with a learning algorihtm. Is there a way to adapt their min C_q bound/procedure for the present case? - more comments on the results of section 5 would be welcomed. Typos Let Q^* minimize (without S) at everY node l. 93 weird parenthesis UPDATE: rebuttal read. I am satisfied with the answers provided by the authors to all the reviews.

Reviewer 3



Paper describes an approach for analyzing of the generalization performance of the subtrees of classification decision tree and evaluates it empirically against the well-known pruning approach. The strong points of the paper: + The paper addresses an interesting question of decision trees regularization techniques and proposes what seems to be novel strategy of choosing a subtrees by using PAC-Bayesian framework. + Experimental section, where the proposed approach, based on the dyadic and kd trees is compared to the pruning method of regularizing of decision trees. + Theoretical contribution is solid and seems to be correct. The weak points of the paper: — Generally, I found the paper quite convoluted and partially hard to follow due to some reasons. Several points should be specified: in line 68-69 authors assume that the diameter of set is $1$, which is possible when underlying norm is the supremum norm, but do not specify the latter. Also, both term “leaves” and “nodes” are used in the paper, which is some place is confusing. In line 108-109 the constant C is defined, however precise meaning of it is not discussed (for example one would be interested how C scales with dimension). — Definition 2 could be presented in more formal way, introducing formal (mathematical) notion of hierarchical partition and trees elements. — Typo: at figure 1 fof —> of — Generally the paper was hard to understand and I tend to reject it for clarity and coherence reasons. However, I am absolutely not an expert in the field of PAC-Bayes learning. ============================ UPDATE: I thank authors for addressing my questions in their rebuttal. I have changed my score towards acceptance.